# Risk factors for neonatal mortality: an observational cohort study in Sarlahi district of rural southern Nepal

Tingting Yan ,[1] Luke C Mullany,[1] Seema Subedi ,[1] Elizabeth A Hazel ,[1] Subarna K Khatry,[1,2] Diwakar Mohan,[1] Scott Zeger,[3] James M Tielsch,[4] Steven C LeClerq,[1,2] Joanne Katz[1]

[1]Department of International Health, Johns Hopkins University Bloomberg School of Public Health, Baltimore, Maryland, USA
[2]Nepal Nutrition Intervention Project - Sarlahi (NNIPS), Nepal Eye Hospital Complex, Tripureshwor, Kathmandu, Nepal
[3]Department of Biostatistics, Johns Hopkins University Bloomberg School of Public Health, Baltimore, Maryland, USA
[4]Department of Global Health, George Washington University School of Public Health and Health Services, Washington, DC, USA

**Correspondence to**
Dr Joanne Katz; jkatz1@jhu.edu

## ABSTRACT

**Objectives** To assess the association between maternal characteristics, adverse birth outcomes (small-for-gestational-age (SGA) and/or preterm) and neonatal mortality in rural Nepal.

**Design** This is a secondary observational analysis to identify risk factors for neonatal mortality, using data from a randomised trial to assess the impact of newborn massage with different oils on neonatal mortality in Sarlahi district, Nepal.

**Setting** Rural Sarlahi district, Nepal.

**Participants** 40 119 pregnant women enrolled from 9 September 2010 to 16 January 2017.

**Main outcome** The outcome variable is neonatal death. Cox regression was used to estimate adjusted Hazard Ratios (aHRs) to assess the association between adverse birth outcomes and neonatal mortality.

**Results** There were 32 004 live births and 998 neonatal deaths. SGA and/or preterm birth was strongly associated with increased neonatal mortality: SGA and preterm (aHR: 7.09, 95% CI: (4.44 to 11.31)), SGA and term/post-term (aHR: 2.12, 95% CI: (1.58 to 2.86)), appropriate-for-gestational-age/large-for-gestational-age and preterm (aHR: 3.23, 95% CI: (2.30 to 4.54)). Neonatal mortality was increased with a history of prior child deaths (aHR: 1.53, 95% CI: (1.24 to 1.87)), being a twin or triplet (aHR: 5.64, 95% CI: (4.25 to 7.48)), births at health posts/clinics or in hospital (aHR: 1.34, 95% CI: (1.13 to 1.58)) and on the way to facilities or outdoors (aHR: 2.26, 95% CI: (1.57 to 3.26)). Risk was lower with increasing maternal height from <145 cm to 145–150 cm (aHR: 0.78, 95% CI: (0.65 to 0.94)) to ≥150 cm (aHR: 0.57, 95% CI: (0.47 to 0.68)), four or more antenatal care (ANC) visits (aHR: 0.67, 95% CI: (0.53 to 0.86)) and education >5 years (aHR: 0.75, 95% CI: (0.62 to 0.92)).

**Conclusion** SGA and/or preterm birth are strongly associated with increased neonatal mortality. To reduce neonatal mortality, interventions that prevent SGA and preterm births by promoting ANC and facility delivery, and care of high-risk infants after birth should be tested.

**Trial registration number** NCT01177111.

## STRENGTHS AND LIMITATIONS OF THIS STUDY

⇒ A large number of pregnancies followed with frequent visits in the neonatal period ensure a statistically powered analysis.
⇒ The availability of a large amount of data for each participant enables a wide range of risk factor to be investigated.
⇒ The date of last menstrual period was obtained early in pregnancy, limiting recall bias, and stillbirths were identified, but both could lead to some misclassification of preterm birth and neonatal death.
⇒ Missing weights due to early newborn deaths or weighing of the baby 72 hours or more after birth were estimated by using multiple imputation, which increased the variability but reduced bias of risk estimates.

births.[1] Almost 99% of neonatal deaths occur in low- and middle-income countries (LMICs).[2] Neonatal mortality in Nepal declined from 50 per 1000 live births in 1996 to 21 in 2022,[2] but neonatal deaths account for a higher percentage of under-5 child deaths since mortality rates among older children have decreased faster than neonatal mortality.[2,3] If the trend continues, it would take another 50 years[4] to reach the Sustainable Development Goals target of reducing neonatal mortality to 12 per 1000 live births by 2030.[5]

Although Nepal has continuously reformed its primary healthcare system and expanded the number of health facilities,[6] there are still daunting obstacles hindering neonatal health, including lacking access to antenatal care (ANC), limited health infrastructure, poor transport and communication, inadequate affordability[7] and significant discrepancy of Nepalese households' access to health services between urban and rural areas.[8] Studies have also shown numerous risk factors contributing to neonatal mortality.[9–15] Small-for-gestational-age (SGA) and preterm births

## INTRODUCTION

Globally, 2.3 million children died in the first 28 days of life in 2021, with the neonatal mortality rate of 18 deaths per 1000 live

are associated with neonatal deaths in LMICs.[10] Other factors include maternal age[11] and education,[12] household income,[13] previous pregnancy history,[14] tetanus vaccination,[15] attendance at ANC[16] and place of delivery.[17]

To identify relatively important ones and explore more evidence for interventions to reduce neonatal mortality in specific settings, we used data from the Nepal Oil Massage Study (NOMS), a community-based, cluster-randomised controlled trial in rural southern Nepal to conduct an observational analysis to assess the association between maternal characteristics, adverse birth outcomes (SGA and/or preterm) and other maternal and infant characteristics and neonatal mortality.

## METHODS

### Study setting

This study was conducted in Sarlahi district of Southern Nepal, a poor rural area of subsistence farming in the low-lying plains of Nepal bordering Bihar, India. In Sarlahi district, neonatal mortality ranges from 30.0 to 41.9 deaths per 1000 live births, even higher than the average in Nepal[18]; 30% of births are <2500 g and over 92% of births occur at home. ANC in this setting is limited.[19]

### Study design

The NOMS was a cluster-randomised, community-based trial (Clinicaltrials.gov (NCT01177111)). The geographic area was divided into sectors that were randomised to receive promotion of full-body massage with mustard oil, which is the standard practice, or the intervention group that received promotion of full-body massage with sunflower seed oil. All women of childbearing age in the 34 participating Village Development Committees were eligible for the trial. Women who consented to pregnancy surveillance were visited every 5 weeks and asked whether they had their period since the last visit. If not, they were offered a pregnancy test. If positive, they were enrolled in the trial. Women were followed with monthly visits until delivery. A birth assessment was conducted on the first visit to the newborn baby (the day of birth or as soon after birth as possible), and neonates were followed up on 1, 3, 7, 10, 14, 21 and 28 days of age. Vital status was collected on mothers and infants at each of these time points. A verbal autopsy was conducted on the causes of neonatal death and the date of death was also recorded. For the analysis of risk factors for neonatal mortality, only live born infants and their mothers were included.

### Data collection

Data for the trial were collected from November 2010 through January 2017. All data collectors were employed and trained by the study. This included local village women who collected data from the pregnant women in their communities. Data such as anthropometry were collected by study employees trained and standardised to collect these measurements. For more complex survey data and supervisory staff, local people with prior experience

collecting these types of data were employed. At the time women were identified as pregnant and enrolled, date of last menstrual period, maternal age, height, education and reproductive history were recorded. Information about household assets, caste and ethnicity of the family was also collected at the time of enrolment. During pregnancy, baseline and monthly visits in pregnancy recorded receipt of tetanus vaccination, tobacco and alcohol use and maternal morbidity. In the first visit within 72 hours after birth, late pregnancy morbidity, labour and delivery characteristics and immediate newborn care practices were collected. Neonatal characteristics included sex, weight and the time since birth at which weight was measured. The aim was to measure weight as soon after birth as possible.

### Patient and public involvement

Our study was an institutional review board-approved secondary analysis of a pre-existing dataset generated from the NOMS trial. Therefore, it was not appropriate to involve patients or the public in the design, conduct, reporting, or dissemination plans for this research.

### Definitions of variables

Neonatal death was defined as the death of a live-born baby in the first 28 days of life and the neonatal mortality rate was defined as the number of neonatal deaths per 1000 live births.[20] Low birth weight (LBW) was defined as a birth weight less than 2500 g.[21] Birth weight was also divided into four groups: <1500 g as very low birth weight (VLBW), 1500–<2500 g as moderately LBW, 2500–4000 g (4000 g included) as normal birth weight (NBW), >4000 g as high birth weight.[22] For our analysis, it was also categorised into two groups: LBW (<2500 g) and NBW (≥2500 g). Gestational age was calculated by taking the number of days between pregnancy outcome and date of last menstrual period obtained by recall at enrolment in the trial.[23] Preterm births were babies born alive before 37 completed weeks gestation.[24] Gestational age was also categorised into four groups: <32 weeks, 32 to <37 weeks, 37 to <42 weeks and 42–45 weeks, and labelled as very preterm, moderate to late preterm, term and post-term, respectively.[25 26] For our analysis, it was also categorised into two groups: preterm (<37 weeks) and term (37–45 weeks). SGA was defined as newborns with weight below the 10th percentile of newborns using the Intergrowth reference population.[27] Appropriate-for-gestational-age (AGA) was defined as weight between the 10th and 90th percentiles of the reference population, and large-for-gestational-age (LGA) was defined as a weight above the 90th percentile of the reference population.[27] Parity was defined as the number of prior pregnancies resulting in a live or stillbirth. Gravidity was the number of prior pregnancies, regardless of the outcome of the pregnancy.[28]

### Data analysis

The data described above were managed and analysed using Stata V.16.0.[29] Cumulative mortality curves

(the inverse of Kaplan-Meier survival curves) (cumulative mortality) were used to describe the differences in neonatal mortality by SGA and preterm status specifically for four groups: (SGA and preterm, SGA and term/postterm, AGA/LGA and preterm, and AGA/LGA and term/post-term).

Maternal characteristics included demographics, socioeconomic status, lifestyle and reproductive history. Maternal age in pregnancy was categorised as ≤18 years, 18 years to 35 years and >35 years. Maternal height was categorised as <145 cm, 145 cm to <150 cm and ≥150 cm. Socioeconomic information included years of education (no schooling as reference, 1 year to ≤5 years and >5 years), and wealth quintiles based on a standardised score of the total number of household assets including land, animal, transport and mobile phone ownership[30] and caste of the family (Brahmin and Chhetri, Vaishya, Shudra and Muslim). Reproductive information included parity (prior pregnancy but no parity, no prior pregnancy, 1–4 and ≥5), gravidity (0 and ≥1), last pregnancy outcome (no prior pregnancy, at least one live birth, stillbirth, or miscarriage or abortion), a history of child deaths, any prior pregnancy that ended in stillbirth or miscarriage, and if the current pregnancy was a multiple birth (twins or triplets). Maternal exposures in pregnancy included self-reported tobacco and/or alcohol use.

For healthcare in pregnancy and characteristics of delivery, tetanus vaccination receipt, place of delivery (at home or maternal home, at a health post/clinic or in hospital, or on the way to the facility or outdoors) and the number of ANC visits (0, 1, 2 or 3, ≥4) were examined.

Neonatal characteristics included birth weight (taken within 72 hours of birth), gestational age, sex, size for gestational age and singleton/twin/triplet in our analysis. Because weights were taken at varying times after birth, and birth weight was missing for some infants who died very soon after birth or was measured more than 72 hours after birth, we multiply imputed birth weights (and hence SGA percentiles) for those with missing weights (601/3957, 15.2% missing) and also imputed weight at time zero for all those with weights and times of weight measurement. This was done using an empirical Bayes regression model of early neonatal weight change by estimating, then recalibrating from the conditional distribution of each child's birth weight given a single measurement at a known later time or imputing given missing weight based on longitudinal daily weights from day of birth through 10 days in a population of infants from the same study area as this trial.[31]

We did not include the intervention as a covariate in the regression model because the results of the randomised trial are being written up for publication but are not yet citable. However, there was no significant difference in mortality between the intervention and control groups. Hence, not including this variable in the regression should not impact the results presented here.

For estimating HRs, we used Cox regression. The outcome was defined as time from birth to death or time the baby was last seen alive. This analysis takes account of loss to follow-up and variable ages at death.

In the simple Cox regression models, we calculated the crude HR (cHR) and their 95% CI for each covariate. In multivariable regression, we did not include some variables because of collinearity. For example, we did not include both parity and gravidity. We did not include prior miscarriage or stillbirth because we included prior child deaths. We developed four different models to examine the adjusted HRs (aHRs) and their 95% CIs for risk factors. In model 1, we included maternal age, height, years of education, wealth quintile, caste, parity, prior child deaths, tetanus vaccination, place of delivery, number of ANC visits, sex of the infant, preterm, adverse birth outcomes (SGA and preterm, SGA and term/post-term, AGA/LGA and preterm, and AGA/LGA and term/post-term) and singleton/twin/triplet. In model 2, we included all the variables in model 1, except place of delivery and number of ANC visits. We ran the models with and without these two variables because about 10% of the population was missing these variables and we wanted to see whether excluding these variables produced similar regression results for the predictors of interest to when they were included. For models 3 and 4 (in supplemental materials), we included gestational age and size for gestation separately, to compare with models 1 and 2, respectively. We ran each of these four models twice, one time where we used weights and associated SGA/preterm taken within 72 hours of birth, and again using the multiply imputed weights and imputed SGA. This allowed us to compare how the imputation impacted the regression coefficients. Since some women had more than one pregnancy included in the analysis, we accounted for correlation within women using robust variance estimation.[32]

For multivariable Cox regression, we have assessed the fitness of each of the four models using Stata and found that none indicated a significant deviation from the proportional hazard assumption.

## RESULTS

A total of 32 004 live-born babies were included in this analysis. Figure 1 shows the number of pregnancies identified, enrolled and their outcomes (not including imputed birth weight). The baseline characteristics, neonatal mortality and cHR of each potential risk factor are shown in table 1. Most pregnant women were between ages 18 years and 35 years. Fifteen per cent of women were shorter than 145 cm and 67% had no schooling. Twenty-nine per cent had no prior pregnancy and almost no women reported the use of alcohol or tobacco during pregnancy (table 1). While 42% of women delivered in a facility, 97% of births were vaginal deliveries.

Based on the cHRs, maternal height, education, wealth quintile, caste, parity, gravidity, prior child deaths and prior adverse pregnancy outcomes, place of delivery, number of ANC visits, multiple births, birth weight, SGA

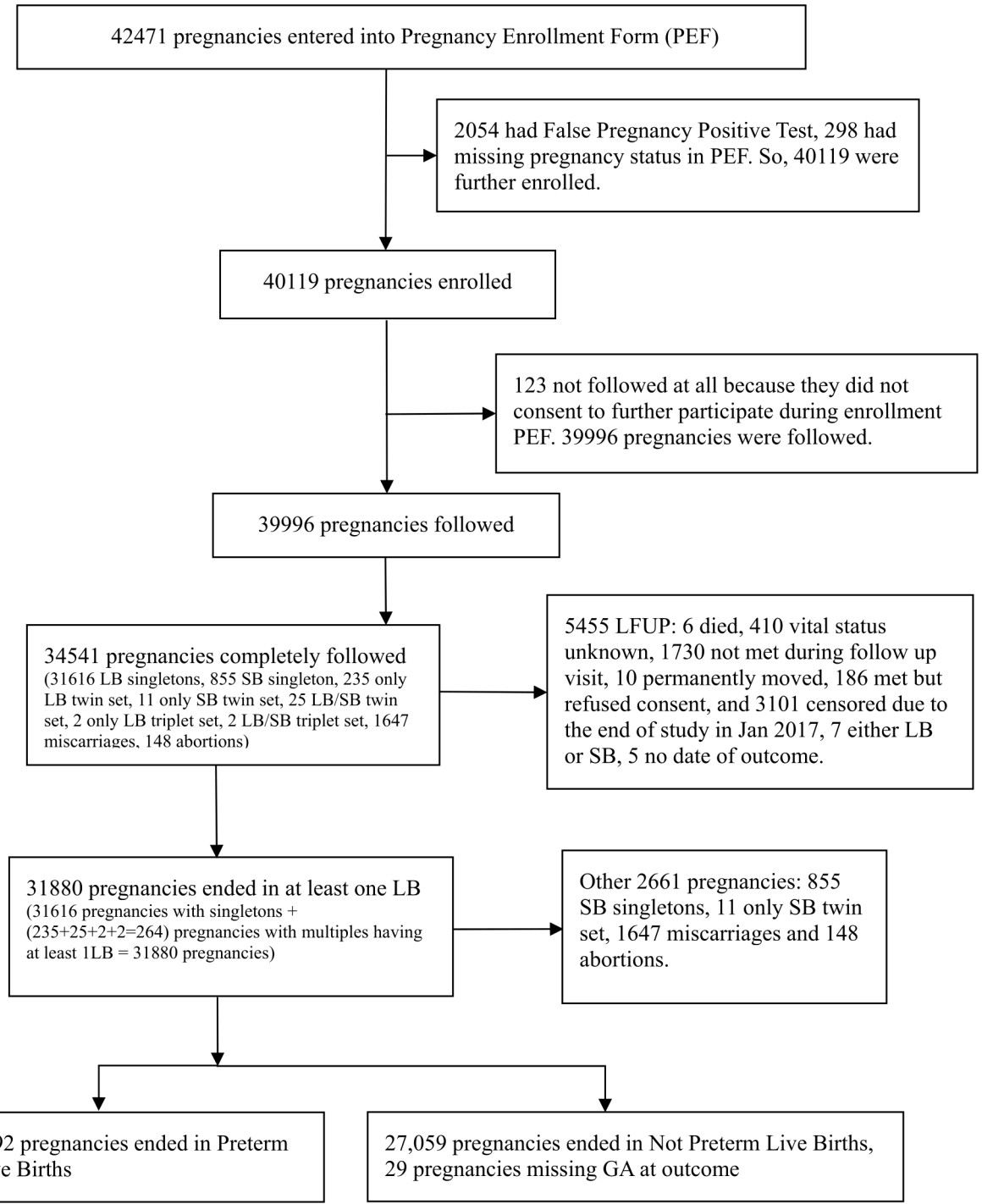

**Figure 1** Flow diagram for participants in Nepal Oil Massage Study. *Data collection was halted for 6 weeks in December 2016. LB, Livebirth; SB, Stillbirth; GA, Gestational Age; LFUP, Lost to Follow Up.

and preterm were all significantly associated with neonatal mortality. Birth weight imputation altered the cHRs for high birth weight (>4000 g) from being a risk factor to being protective and LGA (>90th percentile) from no association to being a significant risk factor. While being SGA and preterm had a large and significant HR in the analysis without imputed weight values, this HR increased significantly when birth weights were imputed.

Variables with high proportions of missing data (around 10%) were ANC visits (10.7%), place of delivery (10.6%),

birth weight (12.4%) and weights taken after 72 hours (13.5%). Neonatal mortality was higher among women who delivered at a facility or on their way to a facility compared with home births. Women who attended four or more ANC visits had lower neonatal mortality risk. Both before and after imputation, VLBW and LBW significantly increased neonatal mortality. Preterm or post-term resulted in higher neonatal mortality. SGA and LGA babies were at higher risk compared with AGA ones. Alcohol and tobacco use was not included in

**Table 1** Characteristics, neonatal mortality and crude HRs of potential risk factors (n=32 004)

| Variables | Study population (n=32 004) | | | Neonatal mortality (per 1000) | Crude HR | 95% CI |
|---|---|---|---|---|---|---|
| | Deaths | Total | Distribution % | | | |
| **Maternal age (years) at LMP** | | | | | | |
| Mean, 95% CI: 22.46 (22.41 to 22.51) | | | | | | |
| ≤18 | 218 | 4948 | 15.46 | 44.06 | 1.57 | (1.34 to 1.83) |
| 18–35 | 748 | 26351 | 82.34 | 28.39 | 1.00 | Reference |
| >35 | 32 | 703 | 2.20 | 45.52 | 1.62 | (1.14 to 2.31) |
| Missing | 0 | 2 | | 0.00 | | |
| **Maternal height (cm)** | | | | | | |
| Mean, 95% CI: 150.55 (150.49 to 150.61) | | | | | | |
| <145 | 232 | 4703 | 14.72 | 49.33 | 1.00 | Reference |
| 145–150 | 332 | 9587 | 30.00 | 34.64 | 0.70 | (0.59 to 0.83) |
| ≥150 | 429 | 17664 | 55.28 | 24.29 | 0.49 | (0.41 to 0.57) |
| Missing | 5 | 50 | | 100.00 | | |
| **Maternal education (years)** | | | | | | |
| Mean, 95% CI: 2.61 (2.56 to 2.65) | | | | | | |
| No schooling | 748 | 21557 | 67.42 | 34.70 | 1.00 | Reference |
| 1–5 | 77 | 2721 | 8.51 | 28.30 | 0.81 | (0.64 to 1.03) |
| >5 | 173 | 7695 | 24.07 | 22.48 | 0.64 | (0.54 to 0.76) |
| Missing | 0 | 31 | | 0.00 | | |
| **Wealth quintile** | | | | | | |
| Poorest | 262 | 6534 | 20.43 | 40.10 | 1.00 | Reference |
| Poorer | 223 | 6407 | 20.03 | 34.81 | 0.87 | (0.72 to 1.05) |
| Middle | 206 | 6400 | 20.01 | 32.19 | 0.80 | (0.66 to 0.97) |
| Richer | 154 | 6303 | 19.71 | 24.43 | 0.60 | (0.49 to 0.74) |
| Richest | 152 | 6339 | 19.82 | 23.98 | 0.59 | (0.48 to 0.73) |
| Missing | 1 | 21 | | 47.62 | | |
| **Caste of the family** | | | | | | |
| Brahmin and Chhetri | 26 | 965 | 3.02 | 26.94 | 1.00 | Reference |
| Vaishya | 682 | 23062 | 72.13 | 29.57 | 1.10 | (0.74 to 1.63) |
| Shudra | 208 | 4947 | 15.47 | 42.05 | 1.57 | (1.05 to 2.37) |
| Muslim and others | 81 | 2999 | 9.38 | 27.01 | 1.01 | (0.65 to 1.57) |
| Missing | 1 | 31 | | 32.26 | | |
| **Parity** | | | | | | |
| 1–4 | 523 | 20444 | 64.22 | 25.58 | 1.00 | Reference |
| ≥5 | 62 | 1398 | 4.39 | 44.35 | 1.75 | (1.33 to 2.30) |
| Prior pregnancy but no parity | 35 | 791 | 2.48 | 44.25 | 1.75 | (1.24 to 2.47) |
| No prior pregnancy | 369 | 9199 | 28.90 | 40.11 | 1.58 | (1.38 to 1.82) |
| Missing | 9 | 172 | | 52.33 | | |
| **Gravidity** | | | | | | |
| ≥1 | 629 | 22804 | 71.26 | 27.58 | 1.00 | Reference |
| 0 (First pregnancy) | 369 | 9199 | 28.74 | 40.11 | 1.46 | (1.28 to 1.67) |
| Missing | 0 | 1 | | 0.00 | | |
| **Last pregnancy outcome** | | | | | | |
| No prior pregnancy | 369 | 9199 | 28.76 | 40.11 | 1.53 | (1.34 to 1.76) |

Continued

**Table 1** Continued

| Variables | Study population (n=32 004) | | | Neonatal mortality (per 1000) | Crude HR | 95% CI |
|---|---|---|---|---|---|---|
| | Deaths | Total | Distribution % | | | |
| At least one live birth | 535 | 20 284 | 63.43 | 26.38 | 1.00 | Reference |
| Stillbirth | 19 | 532 | 1.66 | 35.71 | 1.36 | (0.86 to 2.14) |
| Miscarriage and abortion | 75 | 1965 | 6.14 | 38.17 | 1.46 | (1.13 to 1.89) |
| Missing | 0 | 24 | | 0.00 | | |
| Prior child deaths | | | | | | |
| No prior pregnancy | 369 | 9199 | 29.19 | 40.11 | 1.76 | (1.52 to 2.04) |
| Prior live births but no child deaths | 405 | 17 597 | 55.83 | 23.02 | 1.00 | Reference |
| Prior live births and child death | 149 | 3641 | 11.55 | 40.92 | 1.80 | (1.48 to 2.18) |
| Prior pregnancy but no live birth | 45 | 1081 | 3.43 | 41.63 | 1.83 | (1.33 to 2.53) |
| Missing | 30 | 486 | | 61.73 | | |
| Any prior stillbirth | | | | | | |
| No prior pregnancy | 369 | 9199 | 28.76 | 40.11 | 1.52 | (1.32 to 1.74) |
| Prior pregnancy but no still births | 571 | 21 394 | 66.88 | 26.69 | 1.00 | Reference |
| Prior still births | 58 | 1394 | 4.36 | 41.61 | 1.58 | (1.18 to 2.10) |
| Missing | 0 | 17 | | 0.00 | | |
| Any prior miscarriage | | | | | | |
| No prior pregnancy | 369 | 9199 | 28.75 | 40.11 | 1.54 | (1.34 to 1.77) |
| Prior pregnancy but no miscarriage | 503 | 19 142 | 59.83 | 26.28 | 1.00 | Reference |
| Prior miscarriage | 126 | 3652 | 11.41 | 34.50 | 1.32 | (1.08 to 1.62) |
| Missing | 0 | 11 | | 0.00 | | |
| Any prior multiple births | | | | | | |
| No prior pregnancy | 369 | 9199 | 28.76 | 40.11 | 1.49 | (1.30 to 1.70) |
| Prior pregnancy but no multiples | 611 | 22 484 | 70.30 | 27.17 | 1.00 | Reference |
| Prior multiples | 18 | 299 | 0.93 | 60.20 | 2.25 | (1.38 to 3.68) |
| Missing | 0 | 22 | | 0.00 | | |
| Tetanus vaccination in the past 2 years | | | | | | |
| No | 193 | 5066 | 15.83 | 38.10 | 1.00 | Reference |
| Yes | 805 | 26 938 | 84.17 | 29.88 | 0.78 | (0.66 to 0.92) |
| Missing | 0 | 0 | | | | |
| Any tobacco use in pregnancy | | | | | | |
| No | 987 | 31 649 | 98.89 | 31.19 | 1.00 | Reference |
| Yes | 11 | 355 | 1.11 | 30.99 | 0.99 | (0.55 to 1.80) |
| Missing | 0 | 0 | | | | |
| Any alcohol use in pregnancy | | | | | | |
| No | 997 | 31 908 | 99.70 | 31.34 | 1.00 | Reference |
| Yes | 1 | 96 | 0.30 | 10.42 | 0.33 | (0.05 to 2.37) |
| Missing | 0 | 0 | | | | |
| Place of delivery | | | | | | |
| At home or Maiti* | 437 | 15 872 | 55.46 | 27.53 | 1.00 | Reference |
| At health post/clinic or in hospital | 405 | 12 130 | 42.38 | 33.39 | 1.22 | (1.06 to 1.40) |
| On the way to facility or outdoors | 40 | 617 | 2.16 | 64.83 | 2.43 | (1.70 to 3.48) |
| Missing | 116 | 3385 | | 34.27 | | |
| Number of ANC visits | | | | | | |

**Table 1** Continued

| Variables | Study population (n=32 004) | | | Neonatal mortality (per 1000) | Crude HR | 95% CI |
|---|---|---|---|---|---|---|
| | Deaths | Total | Distribution % | | | |
| 0 | 184 | 5559 | 19.46 | 33.10 | 1.00 | Reference |
| 1 | 148 | 4173 | 14.61 | 35.47 | 1.07 | (0.86 to 1.34) |
| 2 or 3 | 366 | 9872 | 34.55 | 37.07 | 1.12 | (0.93 to 1.35) |
| ≥4 | 183 | 8967 | 31.38 | 20.41 | 0.61 | (0.50 to 0.76) |
| Missing | 117 | 3433 | | 34.08 | | |
| Intervention | | | | | | |
| Mustard oil | 520 | 16 327 | 51.02 | 31.85 | 1.00 | Reference |
| Sunflower oil | 478 | 15 676 | 48.98 | 30.49 | 0.95 | (0.84 to 1.09) |
| Missing | 0 | 1 | | 0 | | |
| Sex of child | | | | | | |
| Male | 548 | 16 531 | 51.73 | 26.63 | 1.00 | Reference |
| Female | 449 | 15 424 | 48.27 | 29.91 | 0.88 | (0.77 to 0.99) |
| Missing | 1 | 49 | | 20.41 | | |
| Birth weight (within 72 hours) | | | | | | |
| Mean, 95% CI: 2773.19 g (2766.95 to 2779.44) | | | | | | |
| VLBW (<1500 g) | 51 | 128 | 0.46 | 398.44 | 69.88 | (49.58 to 98.50) |
| LBW (1500–2500 g) | 199 | 7859 | 28.02 | 25.32 | 3.43 | (2.76 to 4.26) |
| NBW (2500–4000 g) | 141 | 19 354 | 69.01 | 7.29 | 1.00 | Reference |
| HBW (>4000 g) | 3 | 706 | 2.52 | 4.25 | 0.57 | (0.18 to 1.78) |
| Missing | 601 | 3957 | | 151.88 | | |
| Imputed birth weight | | | | | | |
| Mean, 95% CI: 2774.9 g (2770.1 to 2779.8) | | | | | | |
| VLBW (<1500 g) | 65 | 158 | 0.00 | 412.98 | 25.94 | (19.38 to 34.73) |
| LBW (1500–2500 g) | 491 | 8298 | 26.00 | 59.22 | 3.23 | (2.65 to 3.94) |
| NBW (2500–4000 g) | 437 | 23 355 | 73.17 | 18.69 | 1.00 | Reference |
| HBW (>4000 g) | 4 | 110 | 0.00 | 32.72 | 1.74 | (0.59 to 5.14) |
| Missing | 1 | 83 | | 12.05 | | |
| Birth weight (within 72 hours) | | | | | | |
| LBW (<2500 g) | 250 | 7987 | 28.48 | 31.30 | 4.32 | (3.51 to 5.33) |
| NBW (≥2500 g) | 147 | 20 060 | 71.52 | 7.33 | 1.00 | Reference |
| Missing | 601 | 3957 | | 151.88 | | |
| Imputed birth weight | | | | | | |
| LBW (<2500 g) | 556 | 8456 | 26.49 | 65.77 | 3.59 | (2.99 to 4.32) |
| NBW (≥2500 g) | 440 | 23 465 | 73.51 | 18.76 | 1.00 | Reference |
| Missing | 1 | 83 | | 12.05 | | |
| Gestational age (days) | | | | | | |
| Mean, 95% CI: 275.74 (275.49 to 275.99) | | | | | | |
| Very preterm (<32 weeks) | 193 | 662 | 2.12 | 291.54 | 17.44 | (14.54 to 20.91) |
| Moderate to late preterm (32–37 weeks) | 224 | 4244 | 13.62 | 52.78 | 2.66 | (2.25 to 3.14) |
| Term (37–42 weeks) | 451 | 22 371 | 71.78 | 20.16 | 1.00 | Reference |
| Post-term (42–45 weeks) | 111 | 3887 | 12.47 | 28.56 | 1.42 | (1.16 to 1.75) |

**Table 1** Continued

| Variables | Study population (n=32 004) | | | Neonatal mortality (per 1000) | Crude HR | 95% CI |
|---|---|---|---|---|---|---|
| | Deaths | Total | Distribution % | | | |
| Missing | 0 | 3 | | 0.00 | | |
| Gestational age | | | | | | |
| Preterm (<37 weeks) | 417 | 4906 | 15.74 | 85.00 | 4.12 | (3.60 to 4.71) |
| Term (37–45 weeks) | 562 | 26258 | 84.26 | 21.40 | 1.00 | Reference |
| Missing | 0 | 3 | | 0.00 | | |
| Size for gestational age (within 72 hours) | | | | | | |
| AGA | 140 | 13172 | 47.01 | 10.69 | 1.00 | Reference |
| SGA | 244 | 13060 | 46.61 | 18.54 | 1.76 | (1.42 to 2.18) |
| LGA | 12 | 1789 | 6.38 | 6.83 | 0.63 | (0.35 to 1.12) |
| Missing | 602 | 3983 | | 148.36 | | |
| Imputed size for gestational age | | | | | | |
| AGA | 373 | 16127 | 50.57 | 23.15 | 1.00 | Reference |
| SGA | 471 | 14275 | 44.77 | 32.96 | 1.43 | (1.10 to 1.87) |
| LGA | 147 | 1486 | 4.66 | 98.95 | 4.52 | (3.49 to 5.85) |
| Missing | 7 | 116 | | 60.34 | | |
| SGA/preterm (within 72 hours) | | | | | | |
| AGA/LGA and term/post-term | 70 | 11381 | 41.64 | 6.15 | 1.00 | Reference |
| SGA and term/post-term | 185 | 11859 | 43.39 | 15.60 | 2.55 | (1.93 to 3.37) |
| AGA/LGA and preterm | 81 | 3437 | 12.58 | 23.57 | 3.86 | (2.76 to 5.39) |
| SGA and preterm | 54 | 652 | 2.39 | 82.82 | 13.95 | (9.67 to 20.13) |
| Missing | 608 | 4675 | | 130.05 | | |
| Imputed SGA/preterm | | | | | | |
| AGA/LGA and term/post-term | 189 | 13413 | 43.42 | 14.06 | 1.00 | Reference |
| SGA and term/post-term | 391 | 13607 | 41.32 | 28.76 | 2.06 | (1.50 to 2.83) |
| AGA/LGA and preterm | 324 | 4175 | 13.30 | 77.55 | 5.73 | (4.56 to 7.21) |
| SGA and preterm | 79 | 669 | 1.97 | 118.38 | 8.82 | (5.85 to 13.28) |
| Missing | 15 | 140 | | 60.34 | | |
| Singleton/twin/triplet | | | | | | |
| Singleton | 892 | 31502 | 98.43 | 28.32 | 1.00 | Reference |
| Twin or triplet | 106 | 502 | 1.57 | 211.16 | 8.31 | (6.43 to 10.74) |
| Missing | 0 | 0 | | | | |

*Maiti is the maternal home, where women may go to deliver, especially in the first pregnancy.
AGA, appropriate-for-gestational-age; ANC, antenatal care; HBW, high birth weight; LBW, low birth weight; LGA, large-for-gestational-age; LMP, Last Menstrual Period; NBW, normal birth weight; SGA, small-for-gestational-age; VLBW, very low birth weight.

multivariate models, given the low prevalence of their use in pregnancy.

Multivariable regression results are shown in tables 2 and 3. We calculated cHRs for every category for individual variables. If 95% CIs of cHRs covered 1.00, we did not include the variable in our models. These models did not include gravidity, last pregnancy outcome, any prior pregnancy ending in stillbirth, miscarriage or multiples, and tobacco and alcohol use. Model 1 included place of delivery and number of ANC visits, while model 2 did not.

The total number of live births in the analysis was 26 626 with original data (weights taken within 72 hours), and 27 819 with imputed birth weight and SGA data in model 1 (table 2). Place of delivery had a significant impact on neonatal mortality: the risks were higher both at health posts/clinics or in hospital (aHR: 1.34, 95% CI: (1.13 to 1.58)) and on the way to the facility or outdoors (aHR: 2.26, 95% CI: (1.57 to 3.26)); the number of ANC visits was statistically significant for four or more visits (aHR: 0.67, 95% CI: (0.53 to 0.86)).

**Table 2** Multivariate Cox regression model 1

| Variables | Crude HR | 95% CI | Model 1: maternal age, mother's height, mother's years of education, wealth quintile, caste, parity, prior child deaths, tetanus vaccination, place of delivery, number of ANC visits, sex of child, SGA/preterm, imputed SGA/preterm, singleton/twin+triplet | | | |
|---|---|---|---|---|---|---|
| | | | Adjusted HR (n=26 626) | 95% CI | Adjusted HR (using imputed SGA) (n=27 819) | 95% CI |
| **Maternal age (years) at LMP** | | | | | | |
| Mean, 95% CI: 22.46 (22.41 to 22.51) | | | | | | |
| ≤18 | 1.57 | (1.34 to 1.83) | 1.39 | (1.04 to 1.86) | 1.22 | (1.01 to 1.50) |
| 18–35 | 1.00 | Reference | 1.00 | Reference | 1.00 | Reference |
| >35 | 1.62 | (1.14 to 2.31) | 1.28 | (0.60 to 2.77) | 1.36 | (0.84 to 2.21) |
| **Maternal height (cm)** | | | | | | |
| Mean, 95% CI: 150.55 (150.49 to 150.61) | | | | | | |
| <145 | 1.00 | Reference | 1.00 | Reference | 1.00 | Reference |
| 145–150 | 0.70 | (0.59 to 0.83) | 0.64 | (0.48 to 0.86) | 0.75 | (0.62 to 0.92) |
| 150 | 0.49 | (0.41 to 0.57) | 0.57 | (0.43 to 0.75) | 0.55 | (0.46 to 0.67) |
| **Maternal education (years)** | | | | | | |
| Mean, 95% CI: 2.61 (2.56 to 2.65) | | | | | | |
| No schooling | 1.00 | Reference | 1.00 | Reference | 1.00 | Reference |
| 1–5 | 0.81 | (0.64 to 1.03) | 1.01 | (0.69 to 1.49) | 0.92 | (0.71 to 1.20) |
| >5 | 0.64 | (0.54 to 0.76) | 0.79 | (0.57 to 1.09) | 0.78 | (0.62 to 0.97) |
| **Wealth quintile** | | | | | | |
| Poorest | 1.00 | Reference | 1.00 | Reference | 1.00 | Reference |
| Poorer | 0.87 | (0.72 to 1.05) | 0.85 | (0.61 to 1.17) | 0.92 | (0.74 to 1.13) |
| Middle | 0.80 | (0.66 to 0.97) | 0.96 | (0.69 to 1.33) | 0.95 | (0.76 to 1.19) |
| Richer | 0.60 | (0.49 to 0.74) | 0.73 | (0.50 to 1.06) | 0.82 | (0.64 to 1.05) |
| Richest | 0.59 | (0.48 to 0.73) | 0.89 | (0.60 to 1.32) | 0.91 | (0.70 to 1.18) |
| **Caste of the family** | | | | | | |
| Brahmin and Chhetri | 1.00 | Reference | 1.00 | Reference | 1.00 | Reference |
| Vaishya | 1.10 | (0.74 to 1.63) | 1.29 | (0.57 to 2.91) | 0.86 | (0.55 to 1.34) |
| Shudra | 1.57 | (1.05 to 2.37) | 1.61 | (0.69 to 3.76) | 1.01 | (0.63 to 1.64) |
| Muslim and others | 1.01 | (0.65 to 1.57) | 1.04 | (0.43 to 2.52) | 0.69 | (0.41 to 1.14) |
| **Parity** | | | | | | |
| 1–4 | 1.00 | Reference | 1.00 | Reference | 1.00 | Reference |
| ≥5 | 1.75 | (1.33 to 2.30) | 1.20 | (0.66 to 2.19) | 1.04 | (0.70 to 1.55) |
| Prior pregnancy but no parity | 1.75 | (1.24 to 2.47) | 1.99 | (0.43 to 9.11) | 1.63 | (0.69 to 3.84) |
| No prior pregnancy | 1.58 | (1.38 to 1.82) | 1.43 | (1.06 to 1.92) | 1.73 | (1.42 to 2.12) |
| **Prior child deaths** | | | | | | |
| No prior pregnancy | 1.76 | (1.52 to 2.04) | Omitted | | Omitted | |
| Prior live births but no child deaths | 1.00 | Reference | 1.00 | Reference | 1.00 | Reference |
| Prior live births and child death | 1.80 | (1.48 to 2.18) | 1.11 | (0.79 to 1.55) | 1.56 | (1.26 to 1.95) |

**Table 2** Continued

| Variables | Crude HR | 95% CI | Model 1: maternal age, mother's height, mother's years of education, wealth quintile, caste, parity, prior child deaths, tetanus vaccination, place of delivery, number of ANC visits, sex of child, SGA/preterm, imputed SGA/preterm, singleton/twin+triplet | | | |
| | | | Adjusted HR (n=26 626) | 95% CI | Adjusted HR (using imputed SGA) (n=27 819) | 95% CI |
|---|---|---|---|---|---|---|
| Prior pregnancy but no live birth | 1.83 | (1.33 to 2.53) | 0.71 | (0.17 to 2.93) | 1.28 | (0.57 to 2.88) |
| Tetanus vaccination in the past 2 years | | | | | | |
| No | 1.00 | Reference | 1.00 | Reference | 1.00 | Reference |
| Yes | 0.78 | (0.66 to 0.92) | 0.84 | (0.63 to 1.11) | 0.74 | (0.62 to 0.89) |
| Place of delivery | | | | | | |
| At home or Maiti* | 1.00 | Reference | 1.00 | Reference | 1.00 | Reference |
| At health post/clinic or in hospital | 1.22 | (1.06 to 1.40) | 0.94 | (0.73 to 1.21) | 1.34 | (1.13 to 1.58) |
| On the way to facility or outdoors | 2.43 | (1.70 to 3.48) | 1.42 | (0.72 to 2.81) | 2.26 | (1.57 to 3.26) |
| Number of ANC | | | | | | |
| 0 | 1.00 | Reference | 1.00 | Reference | 1.00 | Reference |
| 1 | 1.07 | (0.86 to 1.34) | 1.03 | (0.73 to 1.47) | 1.07 | (0.85 to 1.34) |
| 2 or 3 | 1.12 | (0.93 to 1.35) | 1.06 | (0.79 to 1.42) | 1.06 | (0.87 to 1.29) |
| ≥4 | 0.61 | (0.50 to 0.76) | 0.77 | (0.54 to 1.11) | 0.67 | (0.53 to 0.86) |
| Sex of child | | | | | | |
| Male | 1.00 | Reference | 1.00 | Reference | 1.00 | Reference |
| Female | 0.88 | (0.77 to 0.99) | 1.16 | (0.94 to 1.43) | 0.92 | (0.80 to 1.06) |
| SGA/preterm (within 72 hours) | | | | | | |
| AGA/LGA and term/post-term | 1.00 | Reference | 1.00 | Reference | | |
| SGA and term/post-term | 2.55 | (1.93 to 3.37) | 2.11 | (1.56 to 2.83) | | |
| AGA/LGA and preterm | 3.86 | (2.76 to 5.39) | 3.11 | (2.22 to 4.35) | | |
| SGA and preterm | 13.95 | (9.67 to 20.13) | 6.90 | (4.33 to 10.97) | | |
| Imputed SGA/preterm | | | | | | |
| AGA/LGA and term/post-term | 1.00 | Reference | | | 1.00 | Reference |
| SGA and term/post-term | 2.06 | (1.50 to 2.83) | | | 1.67 | (1.10 to 2.54) |
| AGA/LGA and preterm | 5.73 | (4.56 to 7.21) | | | 4.26 | (3.20 to 5.66) |
| SGA and preterm | 8.82 | (5.85 to 13.28) | | | 3.92 | (2.18 to 7.07) |
| Singleton/twin/triplet | | | | | | |
| Singleton | 1.00 | Reference | 1.00 | Reference | 1.00 | Reference |
| Twin or triplet | 8.31 | (6.43 to 10.74) | 4.89 | (2.91 to 8.21) | 5.43 | (4.01 to 7.37) |

*Maiti is the maternal home, where women may go to deliver, especially in the first pregnancy.
AGA, appropriate-for-gestational-age; ANC, antenatal care; LGA, large-for-gestational-age; LMP, Last Menstrual Period; SGA, small-for-gestational-age.

In model 2 (table 3), the number of participants included was 26 680 with weights taken within 72 hours, and 31 116 after birth weight imputation. The difference was due to more than 10% missing values for place of delivery and number of ANC visits, which were included in model 1, but not in model 2. Results for models 1 and 2 are quite similar, indicating that exclusion of place of

**Table 3** Multivariate Cox regression model 2

| Variables | Crude HR | 95% CI | Adjusted HR (n=26 680) | 95% CI | Adjusted HR (using imputed SGA) (n=31 116) | 95% CI |
|---|---|---|---|---|---|---|
| | | | Model 2: maternal age, mother's height, mother's years of education, wealth quintile, caste, parity, prior child death, tetanus vaccination, sex of child, SGA/preterm, imputed SGA/preterm, singleton/twin/triplet | | | |
| **Maternal age (years) at LMP** | | | | | | |
| Mean, 95% CI: 22.46 (22.41 to 22.51) | | | | | | |
| ≤18 | 1.57 | (1.34 to 1.83) | 1.41 | (1.05 to 1.88) | 1.15 | (0.95, to 1.38) |
| 18–35 | 1.00 | Reference | 1.00 | Reference | 1.00 | Reference |
| >35 | 1.62 | (1.14 to 2.31) | 1.26 | (0.58 to 2.71) | 1.46 | (0.93 to 2.28) |
| **Maternal height (cm)** | | | | | | |
| Mean, 95% CI: 150.55 (150.49 to 150.61) | | | | | | |
| <145 | 1.00 | Reference | 1.00 | Reference | 1.00 | Reference |
| 145–150 | 0.70 | (0.59 to 0.83) | 0.64 | (0.48 to 0.85) | 0.78 | (0.65 to 0.94) |
| ≥150 | 0.49 | (0.41 to 0.57) | 0.57 | (0.43 to 0.75) | 0.57 | (0.47 to 0.68) |
| **Maternal education (years)** | | | | | | |
| Mean, 95% CI: 2.61 (2.56 to 2.65) | | | | | | |
| No schooling | 1.00 | Reference | 1.00 | Reference | 1.00 | Reference |
| 1–5 | 0.81 | (0.64 to 1.03) | 0.98 | (0.67 to 1.44) | 0.89 | (0.69 to 1.14) |
| >5 | 0.64 | (0.54 to 0.76) | 0.75 | (0.55 to 1.04) | 0.75 | (0.62 to 0.92) |
| **Wealth quintile** | | | | | | |
| Poorest | 1.00 | Reference | 1.00 | Reference | 1.00 | Reference |
| Poorer | 0.87 | (0.72 to 1.05) | 0.84 | (0.61 to 1.17) | 0.96 | (0.79 to 1.18) |
| Middle | 0.80 | (0.66 to 0.97) | 0.94 | (0.68 to 1.30) | 0.94 | (0.76 to 1.16) |
| Richer | 0.60 | (0.49 to 0.74) | 0.71 | (0.49 to 1.03) | 0.82 | (0.66 to 1.04) |
| Richest | 0.59 | (0.48 to 0.73) | 0.87 | (0.59 to 1.29) | 0.92 | (0.72 to 1.17) |
| **Caste of the family** | | | | | | |
| Brahmin and Chhetri | 1.00 | Reference | 1.00 | Reference | 1.00 | Reference |
| Vaishya | 1.10 | (0.74 to 1.63) | 1.34 | (0.59 to 3.03) | 0.86 | (0.57 to 1.31) |
| Shudra | 1.57 | (1.05 to 2.37) | 1.68 | (0.72 to 3.93) | 1.00 | (0.64 to 1.57) |
| Muslim and others | 1.01 | (0.65 to 1.57) | 1.08 | (0.44 to 2.62) | 0.67 | (0.41 to 1.07) |
| **Parity** | | | | | | |
| 1–4 | 1.00 | Reference | 1.00 | Reference | 1.00 | Reference |
| ≥5 | 1.75 | (1.33 to 2.30) | 1.21 | (0.67 to 2.21) | 1.05 | (0.72 to 1.52) |
| Prior pregnancy but no parity | 1.75 | (1.24 to 2.47) | 2.01 | (0.44 to 9.24) | 1.52 | (0.71 to 3.27) |
| No prior pregnancy | 1.58 | (1.38 to 1.82) | 1.39 | (1.05 to 1.85) | 1.79 | (1.50 to 2.15) |
| **Prior child deaths** | | | | | | |
| No prior pregnancy | 1.76 | (1.52 to 2.04) | Omitted | | Omitted | |
| Prior live births but no child death | 1.00 | Reference | 1.00 | Reference | 1.00 | Reference |
| Prior pregnancy but no live births | 1.80 | (1.48 to 2.18) | 0.67 | (0.16 to 2.77) | 1.32 | (0.65 to 2.67) |

Continued

**Table 3** Continued

| Variables | Crude HR | 95% CI | Model 2: maternal age, mother's height, mother's years of education, wealth quintile, caste, parity, prior child death, tetanus vaccination, sex of child, SGA/preterm, imputed SGA/preterm, singleton/twin/triplet | | | |
|---|---|---|---|---|---|---|
| | | | Adjusted HR (n=26 680) | 95% CI | Adjusted HR (using imputed SGA) (n=31 116) | 95% CI |
| Prior live births and child death | 1.83 | (1.33 to 2.53) | 1.11 | (0.79 to 1.56) | 1.53 | (1.24 to 1.87) |
| Tetanus vaccination in the past 2 years | | | | | | |
| No | 1.00 | Reference | 1.00 | Reference | 1.00 | Reference |
| Yes | 0.78 | (0.66 to 0.92) | 0.84 | (0.63 to 1.10) | 0.88 | (0.74 to 1.04) |
| Sex of child | | | | | | |
| Male | 1.00 | Reference | 1.00 | Reference | 1.00 | Reference |
| Female | 0.88 | (0.77 to 0.99) | 1.17 | (0.95 to 1.45) | 0.89 | (0.78 to 1.01) |
| SGA/preterm (within 72 hours) | | | | | | |
| AGA/LGA and term/post-term | 1.00 | Reference | 1.00 | Reference | | |
| SGA and term/post-term | 2.55 | (1.93 to 3.37) | 2.12 | (1.58 to 2.86) | | |
| AGA/LGA and preterm | 3.86 | (2.76 to 5.39) | 3.23 | (2.30 to 4.54) | | |
| SGA and preterm | 13.95 | (9.67 to 20.13) | 7.09 | (4.44 to 11.31) | | |
| Imputed SGA/preterm | | | | | | |
| AGA/LGA and term/post-term | 1.00 | Reference | | | 1.00 | Reference |
| SGA and term/post-term | 2.06 | (1.50 to 2.83) | | | 1.68 | (1.20 to 2.35) |
| AGA/LGA and preterm | 5.73 | (4.56 to 7.21) | | | 4.50 | (3.51 to 5.77) |
| SGA and preterm | 8.82 | (5.85 to 13.28) | | | 4.11 | (3.52 to 6.69) |
| Singleton/twin/triplet | | | | | | |
| Singleton | 1.00 | Reference | 1.00 | Reference | 1.00 | Reference |
| Twin or triplet | 8.31 | (6.43 to 10.74) | 4.76 | (2.84 to 8.00) | 5.64 | (4.25 to 7.48) |

AGA, appropriate-for-gestational-age; LGA, large-for-gestational-age; LMP, Last Menstrual Period; SGA, small-for-gestational-age.

delivery and ANC did not change the associations of the other risk factors with mortality.

After imputation and adjustment, variables such as maternal age, wealth quintile and caste of the family were not significantly associated with neonatal mortality. Increasing maternal height from <145 cm to 145–150 cm (aHR: 0.78, 95% CI: (0.65 to 0.94)) to ≥150 cm (aHR: 0.57, 95% CI: (0.47 to 0.68)) and education to >5 years (aHR: 0.75, 95% CI: (0.62 to 0.92)) remained protective factors. Parity was not significantly associated with neonatal mortality. Prior live births and child death (aHR: 1.53, 95% CI: (1.24 to 1.87)) were associated with higher risk. Tetanus vaccination did not have a significant impact on neonatal mortality. Being a twin or triplet (aHR: 5.64, 95% CI: (4.25 to 7.48)) was also significantly associated with neonatal mortality. The findings were similar in models 1 and 2.

Any SGA and/or preterm was strongly associated with an increased likelihood of neonatal mortality (figure 2): SGA and preterm (aHR: 7.09, 95% CI: (4.44 to 11.31)), SGA and term/post-term (aHR: 2.12, 95% CI: (1.58 to 2.86)), AGA/LGA and preterm (aHR: 3.23, 95% CI: (2.30 to 4.54)). Imputation of birth weights increased the association between AGA/LGA and preterm and mortality but not the other categories of adverse birth outcomes.

We also created two other multivariate regression models, models S1 and S2 as shown in online supplemental tables S1 and S2 in the appendices. These included gestational age and size for gestational age as individual variables.

## DISCUSSION

Nepal was one of the few countries to meet the Millennium Development Goal 4 for child mortality but neonatal

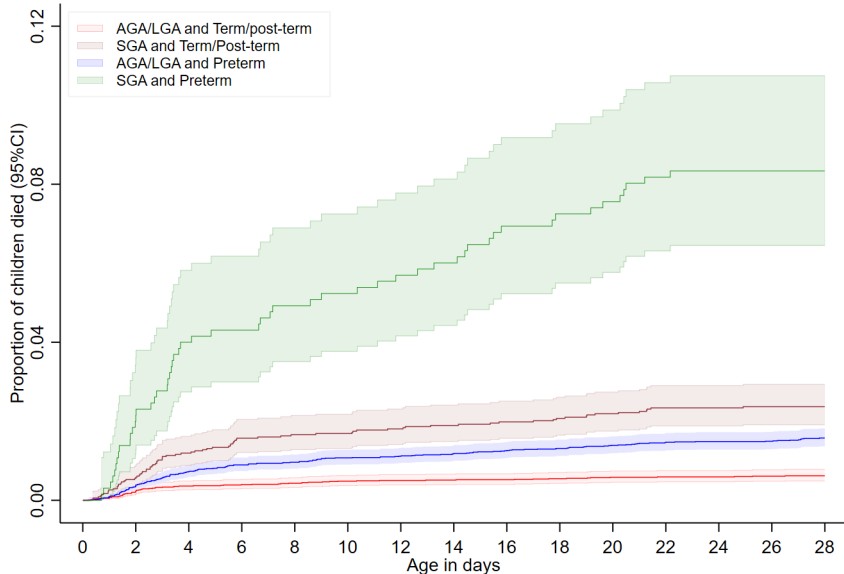

**Figure 2** Cumulative neonatal mortality curves by SGA/preterm excluding imputed data. AGA, appropriate-for-gestational-age; LGA, large-for-gestational-age; SGA, small-for-gestational-age.

mortality rates are still high and need to be reduced by twofold to meet the Sustainable Development Goals of 12 per 1000 live births.[33 34] This study identified maternal characteristics (maternal height, education and prior child death), neonatal characteristics (SGA, preterm and twin or triplet), delivery and healthcare (place of delivery and number of ANC visits) as significant risk factors for neonatal mortality in Sarlahi district, rural southern Nepal.

Compared with previous existing studies, maternal age,[11 35 36] maternal height,[37] maternal education,[12 13] wealth,[13 35] LBW,[36 38 39] SGA and preterm,[36 40–42] twins[13 40 43] and triplets[43] were all strongly associated with neonatal deaths, which correspond to our findings. Previous studies suggest that parity was not considered a predictor of neonatal mortality,[44] which was also supported by our analysis. Healthcare services, specifically the number of ANC[10 16] visits and place of delivery[17] also have an impact on neonatal deaths. Nevertheless, in our study, neonatal mortality was higher among those delivering in a facility, outdoors or on the way to a facility than at home. This is likely due to some women in this population starting their delivery at home, and only going to a facility if something goes wrong during labour and delivery, or being referred to the facility because they were high risk. As the proportion of routine deliveries at facilities increases, this association may reverse.

A previous systematic review indicated that neonatal mortality is reduced by vaccination of women of childbearing age with tetanus toxoid in LMICs,[15] while our study found no association with mortality both before and after adjustment. Neonatal tetanus is an unlikely cause of neonatal deaths due to the high rate of tetanus toxoid vaccination among women of childbearing age in Nepal. A recent programmatic focus on vaccinating adolescent girls with tetanus toxoid may also explain this

lack of association. Tetanus toxoid vaccine receipt during pregnancy (as collected in this study) is often used as a proxy for ANC utilisation in LMICs, but we included the number of ANC visits in the regressions and found it to be associated with neonatal mortality. ANC visits are likely a more proximate risk factor for mortality than tetanus toxoid receipt.

Strengths of this study included a large cohort of pregnancies followed through 28 days postpartum with up to eight visits in the neonatal period. The frequent visits reduced loss to follow-up and improved estimates of age at death. While gestational age was not measured using gold standard ultrasound, the date of last menstrual period was obtained early in the pregnancy, limiting recall bias. However, the reliance on date of last menstrual period could have led to some misclassification of preterm births. Another strength of this study was that stillbirths were identified using maternal recall of whether the baby moved, cried or breathed after birth. If the infant did none of these, they were classified as a stillbirth. However, there could be some misclassification of stillbirths as live births and vice versa. Another limitation is that we did not include maternal morbidity in pregnancy in this analysis. Maternal morbidity may have impacted neonatal mortality but would likely have worked through adverse pregnancy outcomes such as preterm birth and SGA, which were the primary focus of risk factors for mortality in this analysis. Some other limitations include many missing weights due to early newborn death or weighing of the baby 72 hours or more after birth. Most of the missing values for birth weight were estimated by using multiple imputation, which increased the uncertainty of this analysis. The imputation of birth weights also had a qualitative impact on the regression coefficients, in particular for the AGA/LGA and preterm category, which went from an aHR of 3.13 to one of 5.46. This may be due to the fact that

preterm infants who died soon after birth were likely to be missing weights. The neonatal mortality rate among preterm births was 33.02 per 1000 before imputation and 83.19 per 1000 after imputation. Similarly, the neonatal mortality rate among SGA before imputation was 19.10 per 1000 and 32.92 per 1000 after imputation.

In addition to missing birth weights, about 10% of values were missing for place of delivery and number of ANC visits. We also conducted sensitivity analyses including and excluding these covariates and found that the other regression coefficients were not altered significantly by their being included.

Our study found that maternal age, wealth quintile, caste of the family, parity and tetanus vaccination were not significantly associated with neonatal mortality after adjustment and imputation. Although these findings conflict with some of the existing literature, it is possible that the inclusion of more proximate risk factors such as gestational age and size for gestational age lessened the importance of these other more distal risk factors.

## CONCLUSION

Maternal height, maternal education, prior child deaths, SGA, preterm, twins or triplets, place of delivery and number of ANC visits were significantly associated with neonatal mortality risk. Low birthweight babies who are preterm and SGA, preterm and LGA, or SGA and term have different mortality risks and these adverse outcomes likely have different aetiologies. These findings could help refine interventions to reduce these adverse birth outcomes. Interventions that focus on preventing SGA and preterm births, improving education for women, promoting quality antenatal healthcare and facility delivery, and care of vulnerable infants after birth should continue to be emphasised and prioritised.

**Contributors** JK conceived the idea for this paper and is the guarantor for the work. All authors contributed to the planning of the study. JK, TY, LCM, SS, EAH, DM, JMT and SZ had discussions and decided the study design, study methods, data analysis and paper writing. SKK and SCL mainly contributed to the dataset acquisition. EAH was responsible for dataset management, primarily multiple imputation. SS and LCM both contributed a lot to help Stata code writing. TY conducted the analysis, including characteristics description, survival analysis and Cox regression and coordinated the whole research process. SZ, JMT and DM were instrumental in addressing major technical problems related to statistical methods and models construction. All authors contributed to the interpretation of the findings. TY and JK jointly wrote the first draft and revised the manuscript. All authors critically revised the paper for intellectual content and approved the final version of the manuscript.

**Funding** This work was supported by the National Institute for Child Health and Human Development (R01HD092411 and R01HD060712) and the Bill & Melinda Gates Foundation (OPP1084399). The funders did not have a role in the design of the study, the data collection, nor the analysis, interpretation and writing of the manuscript.

**Disclaimer** The views expressed in the submitted article are the authors' own and not an official position of the institution or funder.

**Competing interests** None declared.

**Patient and public involvement** Patients and/or the public were not involved in the design, or conduct, or reporting, or dissemination plans of this research.

**Patient consent for publication** Not applicable.

**Ethics approval** This analysis of secondary data was considered exempt by the Johns Hopkins Bloomberg School of Public Health institutional review board (IRB) (FWA00000287). The original NOMS study received IRB approval from the Johns Hopkins Bloomberg School of Public Health and Tribhuvan University Institute of Medicine in Nepal. Participants gave informed consent to participate in the study before taking part.

**Provenance and peer review** Not commissioned; externally peer reviewed.

**Data availability statement** Data are available upon reasonable request.

**ORCID iDs**
Tingting Yan http://orcid.org/0000-0001-6667-4348
Seema Subedi http://orcid.org/0000-0002-6360-3998
Elizabeth A Hazel http://orcid.org/0000-0002-9176-3278

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
