## [Reviewer comments · BMJ Open]

ARTICLE DETAILS

TITLE (PROVISIONAL)	Risk Factors for Neonatal Mortality; An Observational Cohort Study in Sarlahi District of Rural Southern Nepal
AUTHORS	Yan, Tingting; Mullany, Luke C.; Subedi, Seema; Hazel, Elizabeth; Khatry, Subarna; Mohan, Diwakar; Zeger, Scott; Tielsch, James; LeClerq, Steven; Katz, Joanne

VERSION 1 – REVIEW

REVIEWER	Shiferaw, Kasiye Haramaya University College of Health and Medical Sciences
REVIEW RETURNED	30-Aug-2022

GENERAL COMMENTS	The manuscript addressed one of the challenging indicator in the low and middle income countries. The country has one of the highest neonatal mortality rate in the world. Hence, identifying risks of neonatal mortality may help to tackle the problem. In the following I summarized my comments for authors. Abstract Why objective is not in line with design? I.e. the objective is simply to assess the association of factors and neonatal mortality whereas design introduce the trail that intend to assess impact of newborn massage with different oil on neonatal mortality. Conclusion: what is the implication of your study? Introduction “Each year approximately 2.9 million neonates die worldwide” this data is not updated, please check current report in the world. Previous studies assessed the following according to your introduction: “Small-for-gestational-age (SGA) and preterm births are associated with neonatal deaths in LMICs(6). Other factors include maternal age(7) and education(8), household income(9), previous pregnancy history(10), tetanus vaccination(11), attendance at antenatal care (ANC)(12), and place of delivery(13).” What is the gabs you want to fill using this study? Method Study setting: Would explain more about health infrastructure of the study setting? Study design The design seems you checked effectiveness of promotion of full-body massage with sunflower seed oil. However; you assessed the association of factors with neonatal mortality. How do you negotiate them? “For the analysis of risk factors for neonatal mortality, only live born infants and their mothers were included.” How this is possible excluding outcome variable? What type of sampling technique and procedure did you use? Data collection
--

	Who were data collectors? What are phases of data collection? Or how many times did you collect data during pregnancy to postpartum period? Definition of variables From where you got these variables definitions? Please put citation for your variables definition. "Preterm births were babies born alive before 37 completed weeks gestation." What if born dead at this period? Is not it preterm? How about other variables? The measurement for all variables should be mentioned exhaustively. Data analysis Time to an event was not clearly indicated. Why cox regression was used? Have you checked model fitness? The way you mentioned your model of analysis are not clearly understandable to readers. How you did your variables selection? What type of missing was it? What type of imputation was used? Have you checked multi-collinearity? Is data weighting appropriate here? Result Summarize the characteristic of the study. Do we report crude hazard ratio? Which of the result was reported? Was it imputed or non-imputed? Why? Discussion The main findings were not discussed. It was neither compared with previous studies nor interpreted. What are implication of your studies?
--	--

REVIEWER	Liu, Hao-Tien Chang Gung Memorial Hospital Linkou
REVIEW RETURNED	06-Feb-2023

GENERAL COMMENTS	This authors conducted a study using data from NOMS to analyze the risk factors for neonatal mortality in rural southern Nepal. The work was done quite well, I still have a few comments listed below:  1. In addition to maternal age and height or prior pregnancy outcomes, maternal medical diseases would also influence the neonatal outcomes. Analyzing maternal medical diseases could also help to identify the neonates at risks. Take more care of them could also improve neonatal outcomes. 2. Although most of the participants might have vaginal delivery, 42% of the participants have delivery at health post/clinic or in hospital. How many percentages of participants have Cesarean section for delivery? Does type of delivery influence the neonatal outcomes? 3. Body massage with different kinds of oil were mentioned several times in the method and results. What will this intervention affect neonatal outcomes? Minor issues: 1. There were no full text of the abbreviation, LMICs.
--

VERSION 1 – AUTHOR RESPONSE

Reviewer 1:
 Abstract

Q1: Why objective is not in line with design? I.e. the objective is simply to assess the association of factors and neonatal mortality whereas design introduce the trial that intend to assess impact of newborn massage with different oil on neonatal mortality.

A1: The objective of this secondary observational analysis was to “identify” risk factors for neonatal mortality but the data were generated from a randomized trial. We felt it important to explain the source of the data but the objective of this analysis was different from that of the trial. In the last paragraph of the introduction we explain “We used data from the Nepal Oil Massage Study (NOMS), a community-based, cluster-randomized controlled trial in rural southern Nepal to conduct an observational analysis to assess the association between maternal characteristics, adverse birth outcomes (small-for-gestational-age (SGA) and/or preterm) and other maternal and infant characteristics and neonatal mortality.”

Conclusion

Q2: what is the implication of your study?

A2: From our study, we found that maternal height, maternal education, prior child deaths, SGA, preterm, and twins or triplets, place of delivery and number of ANC visits were significantly associated with neonatal mortality risk in Salahi district, rural southern Nepal. Low birthweight babies who are preterm and SGA, preterm and LGA or SGA and term have different mortality risks and these adverse outcomes likely have different etiologies. These findings could help refine interventions to reduce these adverse birth outcomes. Interventions that focus on preventing SGA and preterm births, improving education for women, promoting quality antenatal health care and facility delivery, and care of vulnerable infants after birth should continue to be emphasized and prioritized. This has been added and emphasized in the conclusions of the study.

Introduction

Q3: “Each year approximately 2.9 million neonates die worldwide” this data is not updated, please check current report in the world.

A3: Thank you. According to UNICEF, we have updated the latest data in 2022. We also described the trends with more explanation in the first paragraph.

Q4: Previous studies assessed the following according to your introduction: “Small-for-gestational-age (SGA) and preterm births are associated with neonatal deaths in LMICs(6). Other factors include maternal age(7) and education(8), household income(9), previous pregnancy history(10), tetanus vaccination(11), attendance at antenatal care (ANC)(12), and place of delivery(13).”

What is the gaps you want to fill using this study?

A4: In low- and middle- income countries, there are various potential risk factors leading to neonatal deaths. However, existing evidence regarding neonatal mortality are limited in rural Nepal. Based on this study, we explored major risk factors and identified relatively important ones focusing on the specific settings in Salahi district, which provided guidance for interventions to reduce neonatal mortality there. For instance, SGA and/or preterm birth are strongly associated with increased neonatal mortality. Other studies examining adverse birth outcomes as risk factors for neonatal mortality have looked at preterm birth primarily, and more recently SGA, but not as many of examined the combination of SGA and/or preterm. We believe these analyses contribute to a growing literature on a more refines categorization of adverse birth outcome and mortality risk that would allow for a more targeted set of interventions appropriate to the etiology of these adverse birth outcomes.

Method

Q5: Study setting: Would explain more about health infrastructure of the study setting?

A5: Yes. Thanks for your suggestion. A description of the health infrastructure is important for our readers to better understand the study settings and objectives of our study. More information about health infrastructure has been provided in the introduction and study settings.

Study design

Q6: The design seems you checked effectiveness of promotion of full-body massage with sunflower seed oil. However; you assessed the association of factors with neonatal mortality. How do you negotiate them?

A6: The Nepal Oil Massage Study (NOMS) was a cluster-randomized, community-based trial, which was designed to determine if sunflower seed oil massage reduced neonatal mortality and morbidity, and improved skin barrier integrity and function. The results of the randomized trial are being written up for publication but are not yet citable. However, there was no significant difference in mortality between the intervention and control groups. Hence we did not include the intervention indicator in the regression model. We have now added this to the methods.

Q7: "For the analysis of risk factors for neonatal mortality, only live born infants and their mothers were included." How this is possible excluding outcome variable?

A7: Thank you. For the analysis, we only included live births, because we explored risk factors for neonatal mortality, which include deaths of live born babies in the first 28 days of life.

Q8: What type of sampling technique and procedure did you use?

A8: In households in 34 Village Development Committees (VDCs) in Sarlahi District of southern Nepal, all women of childbearing age were eligible for enrollment in the pregnancy surveillance. All women who became pregnant, and consented to be in the trial were enrolled and followed through the pregnancy outcome, and through the first 28 days after birth if there was a live born infant. This information is provided in the study design.

Data collection

Q9: Who were data collectors?

A9: All data collectors were employed and trained by the study. This included local village women who collected data from the pregnant women in their communities. More complex data such as anthropometry were collected by study employees trained and standardized to collect these measurements. For more complex survey data and supervisory staff, local people with prior experience collecting these types of data were employed. This has been added to the methods.

Q10: What are phases of data collection? Or how many times did you collect data during pregnancy to postpartum period?

A10: The first phase of the study was pregnancy surveillance. All women of child bearing age were initially identified through a house to house census. They were visited every 5 weeks to see if they had menstruated, and if not, were offered a pregnancy test. If pregnant, those who consented were enrolled in the study. Enrollment in the trial was the second phase of the study. Women were then visited every month during pregnancy until the pregnancy outcome. This was the third phase. The women was visited as soon after birth as possible. This was phase 4. For live born babies, they were visited on days 1, 3, 7, 10, 14, 21 and 28. This was phase 5. This has been described under Study Design in the Methods section.

Definition of variables

Q11: From where you got these variables definitions? Please put citation for your variables definition.

A11: Definitions from these variables were from official data website (e.g. World Health Organization), or the literature. Citation have now been added.

Q12: "Preterm births were babies born alive before 37 completed weeks gestation." What if born dead at this period? Is not it preterm?

A12: Preterm births are defined as live births before 37 completed weeks gestation. Those of a certain gestational age (for example 28 weeks until birth) but who died prior to birth or during labor and delivery are considered stillbirths. They are preterm and stillborn, but we did not include them in our study as we were examining risk factors for neonatal mortality, which does not include stillbirths.

Q13: How about other variables? The measurement for all variables should be mentioned exhaustively.

A13: Every variable and method of measurement has been listed in the methods section and categorization of some of these variables has also been described in the data analysis section.

Data analysis

Q14: Time to an event was not clearly indicated. Why cox regression was used? Have you checked model fitness?

A14: Since not all infants were followed to 28 days (some died and a small number were lost to follow up), we have used time to death as the outcome and hazard ratios as the measure of association with risk factors. Cox regression does specify a proportional hazard ratio. We have assessed the fitness of each of the four models using Stata and found that none indicated a significant deviation from the proportional hazard assumption.

Q15: The way you mentioned your model of analysis are not clearly understandable to readers. How you did your variables selection? What type of missing was it? What type of imputation was used?

A15: Variables were selected based on results from crude hazard ratio (CHR) in table 1, and evidence from the existing literature. We defined reference groups as ones with the highest or lowest likely risk, or previous evidence. Some covariates had so few people in some categories that we did not include them in the multivariable regression (such as tobacco and alcohol use).

Most variables had very few missing values, except for place of delivery and ANC visits (about 10%) and birth weights which occurred either because the infant died prior to be weighed, or was weighed more than 72 hours after birth. The missing place of delivery and ANC visits appears to be missing at random, but those missing birthweights were primarily due to deaths prior to the birth team being able to visit the home to weigh the infant. This missingness was not at random. However, we used multiple imputation with a variety of covariates associated with mortality to impute these birthweights (and provide a citation for the methods used). We present models with and without imputation of birthweights and with and without place of delivery and ANC visits to show any potential for bias associated with missingness.

An explanation was added about the type of imputation. "This was done using an empirical Bayes regression model of early neonatal weight change by estimating, then recalibrating from the conditional distribution of each child's birth weight given a single measurement at a known later time or imputing given missing weight model based on longitudinal daily weights from day of birth through 10 days in a population of infants from the same study area as this trial."

Q16: Have you checked multi-collinearity?

A16: Yes, and we did not include some variables in the multivariable regression because of collinearity. For example, we did not include both parity and gravidity. We did not include prior miscarriage or stillbirth because we included prior child deaths.

Q17: Is data weighting appropriate here?

A17: Apologies but we were not sure what was meant by this question. Do you mean weighting to obtain population based estimates? Or something else? We did not do any weighting of the data.

Result

Q18: Summarize the characteristic of the study.

A18: We have added a summary of study population characteristics based on findings from table 1.

Q19: Do we report crude hazard ratio?

A19: We calculated the crude hazard ratio with 95% CIs for variables as shown in table 1. Crude hazard ratios were used for variables selection. After adjustment by different models, we reported the adjusted hazard ratios as the results.

Q20: Which of the result was reported? Was it imputed or non-imputed? Why?

A20: We reported both unimputed and imputed results in Tables 2 and 3 (Models 1 and 2). The imputed aHRs are in the second column from the right in Tables 2 and 3 and the unimputed are in the 4th column from the right. Similarly for models 3 and 4.

Discussion

Q21: The main findings were not discussed. It was neither compared with previous studies nor interpreted. What are implication of your studies?

A21: We did compare our findings to previous studies. For example, we say in the discussion “Compared with previous existing studies, maternal age(7,21,22), maternal height(23), maternal education(8,9,15), wealth(9,21), low birth weight(22,24,25), SGA and preterm(22,26–28), twins(9,26,29) and triplets(29) were all strongly associated with neonatal deaths, which corresponds to our findings. Previous studies suggest that parity was not considered a predictor of neonatal mortality(30), which was also supported by our analysis. Health care services, specifically the number of ANC(6,12) visits and place of delivery(13) also have an impact on neonatal deaths.”

In terms of the implication, our study found that maternal height, maternal education, prior child deaths, SGA, preterm, and twins or triplets, place of delivery and number of ANC visits were significantly associated with neonatal mortality risk in Salahi district, rural southern Nepal. Low birthweight babies who are preterm and SGA, preterm and LGA or SGA and term have different mortality risks and these adverse outcomes likely have different etiologies. These findings could help refine interventions to reduce these adverse birth outcomes. Interventions that focus on preventing SGA and preterm births, improving education for women, promoting quality antenatal health care and facility delivery, and care of vulnerable infants after birth should continue to be emphasized and prioritized. This is further described in the conclusions.

Reviewer 2:

Q1: In addition to maternal age and height or prior pregnancy outcomes, maternal medical diseases would also influence the neonatal outcomes. Analyzing maternal medical diseases could also help to identify the neonates at risks. Take more care of them could also improve neonatal outcomes.

A1: The reviewer is correct that maternal morbidity in pregnancy may have impacted neonatal mortality but would likely work through impacting adverse pregnancy outcomes such as preterm birth and SGA, which are the primary focus of risk factors for mortality in this analysis. We have added this to the discussion.

Q2: Although most of the participants might have vaginal delivery, 42% of the participants have delivery at health post/clinic or in hospital. How many percentages of participants have Cesarean section for delivery? Does type of delivery influence the neonatal outcomes?

A2: 97% of deliveries were vaginal (3% Caesarean section) so unlikely to influence neonatal mortality significantly. We have added this to the results.

Q3: Body massage with different kinds of oil were mentioned several times in the method and results. What will this intervention affect neonatal outcomes?

A3: The results of the randomized trial are being written up for publication but are not yet citable. However, there was no significant difference in mortality between the intervention and control groups. Hence we did not include the intervention indicator in the regression model. We have now added this explanation to the methods.

Minor issues:

Q4: There were no full text of the abbreviation, LMICs.

A4: Full text of the abbreviation has been added in the paper at the time of first mention.

VERSION 2 – REVIEW

REVIEWER	Liu, Hao-Tien Chang Gung Memorial Hospital Linkou
REVIEW RETURNED	25-Apr-2023
GENERAL COMMENTS	All of the questions I raised have been addressed properly. The authors have also modified their manuscript well according to previous recommendation.

VERSION 2 – AUTHOR RESPONSE